# Eosinophilic Esophagitis and Inflammatory Bowel Disease: What Are the Differences?

**DOI:** 10.3390/ijms25158534

**Published:** 2024-08-05

**Authors:** Hassan Melhem, Jan Hendrik Niess

**Affiliations:** 1Gastroenterology Group, Department of Biomedicine, University of Basel, 4031 Basel, Switzerland; 2Department of Gastroenterology and Hepatology, University Digestive Healthcare Center, Clarunis, 4002 Basel, Switzerland

**Keywords:** inflammatory bowel disease, eosinophilic esophagitis, mucosal barrier integrity, immune–epithelial crosstalk, epithelial barrier disruption, immune-mediated epithelial damage

## Abstract

Eosinophilic esophagitis (EoE) and inflammatory bowel disease (IBD) are chronic inflammatory disorders of the gastrointestinal tract, with EoE predominantly provoked by food and aeroallergens, whereas IBD is driven by a broader spectrum of immunopathological and environmental triggers. This review presents a comprehensive comparison of the pathophysiological and therapeutic strategies for EoE and IBD. We examine the current understanding of their underlying mechanisms, particularly the interplay between environmental factors and genetic susceptibility. A crucial element in both diseases is the integrity of the epithelial barrier, whose disruption plays a central role in their pathogenesis. The involvement of eosinophils, mast cells, B cells, T cells, dendritic cells, macrophages, and their associated cytokines is examined, highlighting the importance of targeting cytokine signaling pathways to modulate immune–epithelial interactions. We propose that advances in computation tools will uncover the significance of G-protein coupled receptors (GPCRs) in connecting immune and epithelial cells, leading to novel therapies for EoE and IBD.

## 1. Introduction

Eosinophilic esophagitis (EoE) and inflammatory bowel disease (IBD) are chronic immune-mediated disorders with significant impacts on patients’ lives. EoE was identified as a distinct disease in the early 1990s after recognizing that eosinophils in the esophagus were not solely linked to gastroesophageal reflux disease [1,2]. Updated guidelines in 2007 formally acknowledged EoE as a chronic, progressive inflammatory disease primarily affecting children and young adults, characterized by symptoms related to esophageal dysfunction and histologically by eosinophil-predominant inflammation [3,4]. Untreated inflammation in EoE can lead to long-term fibrostenotic complications, necessitating complex management strategies such as proton pump inhibitors (PPIs) therapy, swallowed topical steroids, dietary therapy, endoscopic dilation, and biologic agents [5,6]. In contrast, IBD comprises Crohn’s disease (CD) and ulcerative colitis (UC), both marked by chronic intestinal inflammation resulting from dysregulated immune responses to gut microbiota in genetically predisposed individuals [7]. CD can affect any gut segment from the oral cavity to the rectum, often characterized by patchy areas of inflammation that can penetrate multiple layers of the bowel wall, leading to complications such as strictures, fistulas, and abscesses. UC, confined to the colon and rectum, involves continuous inflammation of the mucosal layer, manifesting in symptoms like bloody diarrhea, abdominal cramping, and urgency to defecate [7].

A critical aspect of both diseases is the disruption of the epithelial barrier, which serves as a frontline defense against environmental insults. Genome-wide association studies (GWAS) in both diseases have identified single nucleotide polymorphisms (SNPs) in genes essential for maintaining barrier integrity, pointing towards a disturbed barrier as a prerequisite for the disease. In EoE, the esophageal epithelium becomes permeable due to the inflammatory milieu, allowing allergens and other antigens to penetrate and trigger immune responses [8,9,10]. This increased permeability is facilitated by the reduced tight junction protein expression and the presence of pro-inflammatory cytokines that compromise epithelial integrity [11]. Similarly, in IBD, the intestinal epithelium exhibits increased permeability, often described as a “leaky gut.” This barrier dysfunction permits the translocation of luminal antigens, bacteria, and their products into the underlying tissue, further amplifying the immune response and sustaining chronic inflammation [12,13].

In both EoE and IBD, cytokine production is pivotal in driving inflammation. EoE is predominantly associated with a Th2-mediated immune response characterized by elevated cytokine production, such as IL-4, IL-5, and IL-13 [14]. These cytokines recruit and activate eosinophils, a hallmark of the inflammatory response in EoE [14]. Similarly, Th2 cytokines are also implicated in the pathogenesis of IBD, particularly in UC [15]. However, IBD encompasses a broader spectrum of immune responses, with Th1 and Th17 pathways playing significant roles, especially in CD [15], where anti-IL-23 antibodies have been introduced in clinical treatment.

The immune–epithelial crosstalk plays a pivotal role in the pathogenesis of both EoE and IBD. In EoE, interactions between epithelial cells and immune cells, such as eosinophils and mast cells, generate a persistent inflammatory response that damages the esophageal lining [16]. Similarly, in IBD, the communication between intestinal epithelial cells and immune cells, including macrophages, dendritic cells, and T cells, triggers the release of cytokines and chemokines that sustain the inflammatory cycle [17,18]. This bidirectional dialogue underscores the complexity of these diseases, highlighting the active role of epithelial cells in immune regulation beyond their function as passive barriers.

This review examines the significant parallels between EoE and IBD, focusing on epidemiologic trends, genetic and environmental factors, and therapeutic approaches. We also explore the crucial differences in their immunopathogenesis, emphasizing epithelial-driven mechanisms essential for maintaining barrier function. The similarities in cytokine profiles and epithelial barrier disruption suggest a potential overlap in the immune responses that drive both EoE and IBD. Understanding these shared mechanisms could lead to therapeutic strategies targeting common pathways, offering potential benefits for patients with either condition.

## 2. Increasing EoE and IBD Incidence and Prevalence

Both EoE and IBD have experienced notable increases in incidence and prevalence. The hygiene hypothesis is often cited as a contributing factor, suggesting that reduced exposure to microbes and parasites may lead to these conditions. Similarly, the epithelial barrier hypothesis posits that increased permeability of the epithelial barrier due to exposure to pollutants significantly influences the rising prevalence of both EoE and IBD. Despite these shared mechanistic theories, EoE and IBD predominantly affect different demographic groups and display distinct geographic distributions. EoE is notably the most common cause of dysphagia in young populations in Western regions. In contrast, IBD, historically prevalent in Western countries, is now increasingly diagnosed globally, with substantial incidence and prevalence rates.

### 2.1. EoE

EoE has experienced a notable increase in incidence and prevalence in recent years, a trend not solely due to improved disease recognition [3]. Several population-based studies from the USA [19] and Europe [20,21] indicate that this rise is at least partly occurring and is not due to better recognition through increased awareness of EoE by physicians. Among children undergoing esophagogastroduodenoscopy (EGD) for various reasons, EoE prevalence is 3.7%, and this rate increases significantly to between 63% and 88% when EGD is performed for impaction or dysphagia [22,23] and to 10% to 15% in adults [24]. The latest pooled prevalence data show 34.4 cases per 100,000 inhabitants overall, with a higher rate of 42.2 cases per 100,000 among adults [25]. A comprehensive meta-analysis reports a prevalence of 32.5 per 100,000 in adults and 30.9 per 100,000 in children [26]. A study in Central Spain (2005–2011) found an average annual incidence of 6.3 per 100,000 inhabitants and a prevalence of 44.6 per 100,000 [27]. Many EoE cases have been documented in North America, Western and Eastern Europe, and Australia [28]. There are fewer cases in South America, Asia, and the Middle East [29]. Cases have been reported in Northern Africa, but none in Sub-Saharan Africa or India [30]. In Switzerland’s Canton of Vaud, the prevalence rate in the year 2013 was 24.1 per 100,000 people, and the annual incidence from 2010 to 2013 was 10.6 times higher compared to 1993 to 2009 [31]. Prevalence rates exceeding 100 per 100,000 inhabitants have been reported in some industrialized regions. Overall, EoE is a common disease and not a rare disease as assumed in the past.

### 2.2. IBD

The burden of IBD is increasing worldwide, similar to the trend observed in EoE. The prevalence of IBD varies geographically, reflecting its incidence patterns [32]. IBD is more prevalent in Western countries, affecting 0.2% in Europe [33]. Similarly, a systematic analysis from the Global Burden of Disease Study 2019 identified the United States as having the highest number of cases, with 245.3 cases per 100,000 people [34]. Projections suggest that in 2030, IBD could affect around 4 million individuals in North America [35]. Recent data also show a significant rise in IBD cases in newly industrialized nations, including Asia, Africa, and South America. Whether the increase in IBD in recently industrialized nations will parallel an increase in EoE needs further investigation. For example, Taiwan’s CD prevalence increased from 0.6 to 3.9 per 100,000 and UC from 2.1 to 12.8 per 100,000 between 2001 and 2015 [36]. Similarly, South Korea’s UC prevalence rose from 7.6 per 100,000 in 1997 to 30.9 in 2005 [37,38]. These trends underscore the influence of environmental factors such as urbanization, Western dietary patterns, and lifestyle changes on the disease’s development.

## 3. Genetics Polymorphisms

Although EoE and IBD, particularly UC, invoke Th2-mediated pathways with shared pro-inflammatory cytokines (mainly IL-5 and IL-13) and shared activation of downstream Janus kinase and signal transducer and activator of transcription (JAK-STAT) pathways (mainly STAT3 and STAT6), they have distinct genetic landscapes. Polymorphisms in various genes have been implicated in the pathogenesis of both diseases, highlighting both unique and shared genetic predispositions. Genetic heritability is a low-risk factor for both EoE and IBD, as evidenced by the low odds ratios associated with SNPs in these diseases. This genetic component does not account for the increased incidence and prevalence observed in recent decades, suggesting a significant influence of environmental factors. Although EoE and IBD share specific associated SNPs, EoE patients have only a 3.5-fold increased risk of developing IBD [39], and the prevalence of EoE among IBD patients is merely 1.5% [40]. A population-based prospective cohort analysis revealed a significantly elevated risk of EoE among patients with CD (prevalence ratio [PR] 7.8) or UC (PR 5.0) [41]. This highlights that unique and shared genetic predispositions contribute to the development of EoE and IBD.

### 3.1. EoE

EoE is significantly linked to polymorphisms in genes that mainly affect epithelial barrier integrity and the Th2-mediated immune response, aligning with its underlying pathobiology. Notably, most genetic risk variants for EoE are located outside of gene-coding regions. GWAS have identified genetic variants at five loci associated with EoE, including TSLP, CAPN14, LRRC32/EMSY, CLEC16A/DEX1, and STAT6 [42]. A notable GWAS conducted by Rothenberg et al. identified a significant genetic variant involving the TSLP and WDR36 genes [43]. As TSLP is located on the X chromosome, these findings may explain why males are predisposed to EoE. Kottyan et al. discovered an association with the protease CAPN14 gene, demonstrating its specific expression in the esophageal epithelium, dynamic upregulation by IL-13, and critical role in epithelial homeostasis and repair [44].

Further research by Sleiman et al. reported additional significant genome-wide associations involving four genes: C11orf30, STAT6, ANKRD27, and CAPN14 [42]. Both C11orf30 and STAT6 are linked to various atopic and autoimmune diseases, although analyses indicated that the association of STAT6 with EoE is independent of sensitization status. STAT6 is integral to the IL-4 and IL-13 pathways, crucial for controlling the Th2 immune response. Other genes of interest include those involved in eosinophil recruitment and activation, such as the eotaxin-3 (CCL26) gene, where polymorphisms can lead to increased eosinophilic inflammation in the esophagus [45]. Studies have identified the toll-like receptor 3 (TLR3) as an EoE susceptibility locus, with effects independent of TSLP [46]. These discoveries emphasize the complex genetic architecture of EoE and highlight several pathways and mechanisms contributing to the disease’s pathogenesis.

### 3.2. IBD

In contrast, IBD is characterized by a broader range of genetic polymorphisms, reflecting its complex and multifactorial nature. To date, GWAS have revealed more than 240 risk variants for IBD, many of which are shared between CD and UC, indicating common pathways in their etiology. Notably, genes involved in immune regulation, epithelial barrier function, and microbial interactions are prominently featured among these loci [47,48,49]. For example, NOD2, a gene that recognizes bacterial peptidoglycans, is strongly associated with CD, particularly in individuals of European descent [50,51]. Variants in the IL23R gene, which encodes a subunit of the interleukin-23 receptor, have been linked to both CD and UC, underscoring the importance of the IL-23/Th17 axis in IBD pathogenesis [52]. Mutations in genes such as ATG16L1 and IRGM [53], which are involved in autophagy and the handling of intracellular bacteria, further illustrate the role of genetic factors in modulating immune responses and maintaining intestinal homeostasis.

### 3.3. Overlapping Genetic Features

Despite the distinct genetic landscapes of EoE and IBD, notable overlaps suggest shared pathogenic pathways. Both conditions involve genes related to immune regulation, epithelial barrier function, and cytokine signaling, indicating common mechanisms that may drive chronic gastrointestinal inflammation.

One of the critical overlapping genetic features involves the STAT6 gene, which plays a crucial role in Th2 cell differentiation and signaling. Polymorphisms in STAT6 have been associated with both EoE [42] and IBD [54], highlighting the importance of Th2-mediated immune responses in these diseases. In EoE, STAT6 activation leads to increased expression of eotaxins and other chemokines that recruit eosinophils to the esophagus. In IBD, particularly in UC, STAT6 polymorphisms are associated with an enhanced Th2 response, contributing to mucosal inflammation.

Polymorphisms in the IL-13 gene also illustrate the genetic overlap between EoE [55] and IBD [56]. IL-13 is a cytokine involved in regulating inflammatory responses and epithelial cell function. In EoE, IL-13 overexpression contributes to eosinophil recruitment and esophageal tissue remodeling. In IBD, IL-13 polymorphisms are linked to altered immune responses implicated in CD and UC pathogenesis, mainly in the context of abnormal Th2 responses.

Another shared genetic factor is the filaggrin (FLG) gene, crucial for maintaining epithelial barrier integrity. In EoE, FLG mutations can lead to increased permeability of the esophageal epithelium, allowing allergens and antigens to penetrate and trigger immune responses [57]. Similarly, FLG loss-of-function variants are also associated with IBD but do not affect IBD susceptibility. Similarly, loss-of-function variants in the FLG gene are also associated with IBD. However, these variants do not influence susceptibility to IBD [58]. Understanding these shared genetic polymorphisms provides insights into the common pathways driving these diseases. It highlights potential targets for therapeutic interventions that could benefit patients suffering from either or both conditions.

## 4. The Exposome in EoE and IBD

The environment has a significant role in developing and exacerbating both EoE and IBD, although the specific influences and mechanisms differ between these conditions. In EoE, dietary allergens are primary environmental triggers. Exposure to food antigens such as milk, wheat, soy, fish and shellfish, and nuts can lead to an allergic inflammatory response characterized by eosinophil infiltration in the esophagus [59,60,61]. The seasonal variation in EoE incidence suggests that aeroallergens like pollen may also contribute to disease flares, possibly through cross-reactivity with food allergens [62]. Additionally, antibiotics [63] and alterations in the microbiome have been linked with an increased EoE risk [64]. Detergents, such as sodium dodecyl sulfate in dish soap and toothpaste, compromise esophageal barrier integrity, stimulate IL-33 production, and induce epithelial hyperplasia and tissue eosinophilia [65]. These findings suggest that detergents may play a significant role as environmental triggers in the pathogenesis of EoE.

In contrast, IBD is influenced by a broader range of environmental factors (extensively reviewed in [66]). Diet [67], smoking [68], and the use of medications like nonsteroidal anti-inflammatory drugs are well-known contributors to IBD development and flares [69]. A Western diet with high fat and sugar intake and low fibers has been linked to dysbiosis and increased intestinal permeability, promoting chronic inflammation in the gut [70]. Additionally, antibiotic use and infections in early life can disrupt the gut microbiota, potentially setting the stage for IBD in genetically susceptible individuals [71,72]. Low vitamin D levels provide an elevated risk of developing IBD [73]. In mouse models, vitamin D deficiency increases susceptibility to dextran sodium sulfate-induced colitis. Supplementation with 1.25(OH)2D3 has been demonstrated to mitigate the severity of intestinal inflammation [74]. Interestingly, vitamin D is also linked to EoE. Vitamin D levels are inversely correlated with esophageal eosinophilia and epithelial histopathology severity in a preclinical model of IL-13-mediated esophageal allergic inflammation and human EoE [75].

## 5. Diagnosis

Diagnosing EoE and IBD involves distinct but somewhat overlapping approaches. The assessment of EoE focuses on symptoms such as dysphagia and food impaction. However, the gold standard for EoE diagnosis involves histological examination of esophageal biopsy. Because eosinophilic infiltration in the esophagus can be patchy, it is recommended to take biopsy samples from both the proximal and distal regions of the esophagus to enhance diagnostic accuracy [76]. To maximize sensitivity, at least five biopsy specimens should be collected from the proximal and distal esophagus, adhering to a diagnostic threshold of 15 or more eosinophils per high-power field [3]. Recent research has introduced the EoE histologic severity scoring index. This new histological scoring system considers additional inflammatory features beyond just eosinophil counts, providing a more comprehensive assessment of the disease’s severity and EoE-like entities not associated with esophageal eosinophilia [77].

Diagnosing IBD involves a thorough approach with clinical evaluation, lab tests [78] for inflammation markers (CRP, ESR), fecal calprotectin, imaging (MRI, CT scans, and intestinal bowel ultrasound), endoscopy [79], and histology. It starts with patient history and physical exams to identify symptoms like chronic diarrhea, abdominal pain, and rectal bleeding. Stool tests can rule out infections and measure inflammation through fecal calprotectin or lactoferrin. Ultrasound assesses bowel wall thickness, particularly in children. Endoscopy is critical for visualizing the GI tract and obtaining biopsies. A colonoscopy examines the colon and terminal ileum to diagnose UC and CD. Upper endoscopy and capsule endoscopy are used for suspected upper GI involvement and small intestine visualization in CD. Biopsies confirm the diagnosis by identifying histological features like continuous mucosal inflammation in UC and patchy, transmural inflammation with granulomas in CD. Additionally, the Mayo score is often used to assess the severity of UC, ranging from 0 to 12, based on stool frequency, rectal bleeding, endoscopic findings, and the physician’s global assessment. Combining these diagnostic approaches allows for a comprehensive evaluation of suspected IBD, ensuring accurate diagnosis and appropriate management of the disease.

## 6. Treatments

Treatment approaches for EoE and IBD differ due to the distinct pathophysiology and target tissues involved in each condition (Table 1). In EoE, the mainstay of therapy revolves around dietary modifications and pharmacotherapy aimed at reducing esophageal inflammation and symptoms. Dietary interventions often involve the elimination of specific food triggers through an elimination diet followed by food reintroduction [80]. This approach, known as dietary elimination therapy, can effectively induce remission in many patients with EoE. PPIs [81,82,83] are commonly prescribed to suppress gastric acid secretion, alleviate reflux symptoms, and reduce esophageal inflammation. Swallowed topical corticosteroids [84,85] achieve short-term symptomatic relief and induce mucosal healing, especially in severe cases or during acute flares. Other pharmacological agents, such as leukotriene receptor antagonists, were inefficient in treating EoE [86]. Biologic agents, particularly those targeting IL-13 have been tested [87]. The antibody dupilumab targeting the shared receptor chain of the IL-4/IL-13 receptor has been recently approved for patients with severe or treatment-resistant EoE or intolerance to topical corticosteroids [88]. The management of IBD involves a comprehensive array of treatment strategies tailored to the severity of the disease, its location, and individual patient responses. The primary objectives of IBD treatment are to induce and maintain remission, alleviate symptoms, enhance the quality of life, and prevent complications. Standard medications include aminosalicylates (5-ASAs) for mild-to-moderate cases, corticosteroids for acute flares, and immunomodulators like azathioprine, 6-mercaptopurine, or methotrexate for maintenance therapy to reduce reliance on steroids [89]. Biologic therapies have significantly advanced the treatment of moderate-to-severe IBD. These therapies target specific immune response components, such as tumor necrosis factor-alpha (TNF-α), interleukins, and integrins, providing targeted immunosuppression while minimizing systemic side effects. Anti-TNF agents, anti-integrins, and anti-IL-12/23 or anti-IL-23 agents are among the biologics that have revolutionized IBD management [89,90]. Fecal microbiota transplantation (FMT) has emerged as a promising therapeutic approach to restore diversity within the altered microbiome with reduced diversity in IBD patients [91]. Intensive, multi-donor FMT protocols have shown the greatest efficacy, suggesting that transferring a highly diverse fecal microbiota is crucial for successfully reconstitution microbial communities in IBD [92,93]. In cases where medical therapy is insufficient, surgical interventions such as bowel resection or ostomy may be necessary to address complications like strictures, perforations, or refractory disease unresponsive to conventional treatments.

## 7. The Epithelial Barrier in EoE and IBD

The epithelial barrier in the esophagus and intestine maintains overall health and protects the body from external threats. This barrier, composed of tightly connected epithelial cells, is the first line of defense against pathogens, toxins, and antigens. In the esophagus, the epithelium prevents harmful substances from penetrating deeper tissues, thus protecting against infections and inflammation. Similarly, in the intestine, the epithelial barrier defends against pathogens, regulates nutrient absorption, and maintains gut homeostasis. Dysfunction of the epithelial barrier has been implicated in a spectrum of diseases, ranging from allergic conditions to autoimmune disorders and beyond. Understanding the intricate interplay between epithelial barrier function and disease pathogenesis is essential for unraveling the underlying mechanisms driving these conditions and developing targeted therapeutic interventions.

### 7.1. Epithelial Barrier Dysfunction in EoE

The esophageal epithelium is a stratified squamous epithelium lining the inner surface of the esophagus from the pharynx to the stomach, lacking structures like villi and crypts found in the intestine. It consists of several layers: the basal layer, where undifferentiated basal cells proliferate; the spinous layer, with actively dividing cells forming desmosome junctions; the granular layer, where cells undergo terminal differentiation; and the superficial layer of flattened squamous cells that are continuously shed and replaced [8,94] (Figure 1).

This complex and dynamic barrier maintains tissue homeostasis and protects against external insults, playing a crucial role in EoE [8,10]. Transcriptome analysis in EoE patients has revealed substantial transcriptional changes in approximately 40% of esophagus-specific genes, with nearly 90% of these affected genes being downregulated [42,43,44,45,95,96]. These genes are primarily associated with essential pathways such as keratinization, epidermal development, and cellular differentiation [97]. This extensive alteration highlights the fundamental importance of the epithelial barrier in EoE, as its disruption is closely linked to the disease’s characteristic inflammatory and remodeling processes.

A series of well-coordinated processes govern the formation and maintenance of the esophageal epithelial barrier. Each epithelium layer contains a specific set of proteases and their inhibitors in cellular compartments and the extracellular space. The balance between these proteases and inhibitors is crucial for developing and maintaining the epithelial barrier and sensing damage and environmental insults, enabling immune responses and tissue regeneration [8]. In EoE, this balance is disrupted, leading to mucosal barrier abnormalities. Dysregulated epithelial protease activity is a hallmark of EoE. GWAS have identified CAPN14, an intracellular calcium-activated protease, as highly associated with EoE. CAPN14 is expressed explicitly in the esophagus, and experiments with overexpression and silencing of CAPN14 in cultured esophageal cells have shown significant disruption of epithelial barrier function.

Moreover, serine protease inhibitors (SERPINs) and serine protease inhibitors, Kazal type (SPINKs), are among the most dysregulated peptidase families in EoE. Notably, SPINK5 expression is reduced in EoE, increasing proteolytic activity in esophageal epithelial cells [97]. In addition to SPINK5, SPINK7 is also implicated in EoE, linked to impaired epithelial differentiation, reduced barrier integrity, and heightened proinflammatory responses, mainly through increased production of TSLP [98].

Alarmin cytokines originating from the epithelium, such as TSLP and IL-33, act as danger signals upon tissue damage. They play a crucial role in promoting TH2 immune responses by influencing the development of adaptive responses and directly impacting various allergic effector cells, including eosinophils, mast cells, and basophils [99]. GWAS have implicated genetic variants in TSLP in EoE susceptibility [42,43,44]. Individuals carrying the risk allele for the TSLP variant most associated with EoE show elevated esophageal TSLP RNA expression [43]. TSLP and IL-33 gene expressions are increased in esophageal biopsy specimens from children with EoE. Mice genetically deficient in TSLPR or IL-33R/ST2 show attenuated inflammation in experimental EoE-like disease.

Additionally, IL-33 protein levels are markedly increased within the nuclei of basal layer esophageal epithelial cells in patients with active EoE compared to controls, with levels normalizing upon EoE remission. A new allergen-sensing pathway called RipIL-33 has been discovered in esophageal epithelial cells, revealing how allergenic proteins can stimulate the production of alarmin cytokines. Notably, the RIPK1-caspase 8 ripoptosome complex acts as an allergen sensor, detecting allergic triggers and leading to the activation and release of IL-33 through caspase 8 [100].

The lower layers of the human esophageal squamous epithelium are made up of actively dividing cells. These cells undergo differentiation, becoming flatter as they migrate to the luminal surface, where they connect to form a barrier that protects against repeated exposure to external antigens. An increase in basal cell numbers, known as basal cell hyperplasia, is a well-documented histological change linked to active disease [101]. This condition is associated with higher eosinophil and mast cell numbers in EoE [45]. Basal cell hyperplasia is notably decreased by fluticasone propionate treatment in both the esophagus’s proximal and distal regions [102]. Mice repeatedly exposed to *A. fumigatus* or intratracheal IL-13 have demonstrated that esophageal eosinophilia and basal cell hyperplasia develop through mechanisms reliant on STAT6 and IL-5 [103].

Genes involved in keratinization are among the most downregulated biological processes in the esophagus-specific transcripts altered during EoE [97,104]. Additionally, junctional proteins such as E-cadherin, claudin-1, and desmoglein-1, which are necessary to maintain barrier integrity, are significantly downregulated during EoE [105,106,107]. These changes compromise the barrier function of the esophageal epithelium, facilitating the passage of antigens and subsequent immune activation, which are central to the pathogenesis of EoE.

### 7.2. Epithelial Barrier Dysfunction in IBD

The intestinal epithelium differs significantly from the esophageal epithelium in composition and structure, reflecting their unique roles in the digestive system. The intestinal epithelium consists mainly of columnar epithelial cells arranged into villi and crypts, which increase the surface area for absorption and secretion [108] (Figure 2). These cells are specialized for nutrient absorption, mucus and enzyme secretion, and protection against pathogens and toxins. Goblet cells secrete mucus to lubricate and protect the epithelial lining, while enteroendocrine cells release hormones that regulate digestion and appetite. Paneth cells at the crypt bases secrete antimicrobial peptides to maintain gut microbial balance [109,110]. Unlike the esophagus, which lacks a mucus layer, the intestinal mucus is thicker and aids in lubrication and protection against digestive enzymes and pathogens. These differences underscore the specialized functions and adaptations of the intestine and esophagus within the digestive system.

Barrier dysfunction is a critical factor in IBD pathogenesis, disrupting the balance needed for tissue homeostasis. A hallmark of IBD, “epithelial leakage”, compromises barrier integrity, allowing luminal components to infiltrate and trigger immune responses, leading to inflammation. Evidence suggests barrier defects may precede disease onset, as increased permeability is seen even in remission [111]. Mouse models, including SAMP/YitFc and *Mdr1a^−/−^*, show increased permeability before inflammation [112,113].

Microscopically, IBD patients exhibit reduced goblet cells, thinner mucus layers, and abnormal glycosylation, further weakening the barrier [114]. In *Muc2*-deficient mice, a diminished mucus layer and elevated pro-inflammatory markers are noted [115]. Increased small bowel and colonic permeability in CD correlates with inflammation and predicts relapse. Genetic variants affecting barrier function, such as CARD15/NOD2, HFN4, CDH1, and LAMB1, link to severe IBD forms [116]. Mouse models with tight junction protein alterations, like JAM-a deficient [117] and claudin-2 transgenic mice [118], show that a leaky barrier alone does not induce inflammation, underscoring the complex interplay in IBD pathogenesis. Epithelial changes, such as upregulation of Claudin-2 and downregulation of claudin-5, -8, and occludin, impair barrier function in IBD [119]. UC-specific downregulation of claudin-4, -7, and occludin further compromises integrity [120]. Dysregulation of tight junction proteins, including occludin and tricellulin, and activation of MLCK, leading to MLC phosphorylation and occludin endocytosis, increase permeability [121]. Understanding these epithelial alterations is vital for developing targeted therapies to restore barrier integrity and mucosal homeostasis.

### 7.3. Immune and Epithelial Interfaces: Soluble Immune Effectors

#### 7.3.1. EoE

The intricate interplay between immune cells and epithelial cells is fundamental to the pathogenesis of EoE, as it orchestrates the inflammatory and remodeling processes characteristic of this chronic allergic condition. Epithelial-to-mesenchymal transformation (EMT) involves epithelial cells losing their polarity and adhesion properties, and acquiring mesenchymal characteristics such as motility [122,123]. In EoE, EMT and subepithelial fibrosis may be induced by TGF-β and major basic protein (MBP) released by eosinophils or damaged epithelium [123,124]. The extent of EMT in EoE patients correlates with TGF-β1 levels, eosinophil count, and subepithelial fibrosis. Eosinophils can also trigger the expression of EMT and fibrosis-related factors in epithelial cells, such as TGF-α and MMP-9, via MBP and cytokines like IL-13 [122].

TGF-β promotes esophageal remodeling in EoE by activating fibroblasts and inducing the secretion of extracellular matrix (ECM) proteins like collagen and fibronectin and promoting smooth muscle proliferation, hyperplasia, and contractility [103,125]. TGF-β levels are elevated in EoE esophageal biopsy specimens and are produced by infiltrating eosinophils and mast cells. Moreover, TGF-β1 impairs esophageal epithelial barrier function by reducing claudin-7 levels [126]. More recently, it has been shown that mice carrying a loss-of-function variant of TGFβR1, identified in atopic patients, spontaneously develop a condition that mirrors EoE in clinical, immunological, histological, and transcriptional aspects [127]. The same study has demonstrated that epithelial cells expressing the TGFβR1 variant are hyperproliferative, fail to differentiate correctly, and overproduce innate pro-inflammatory mediators, even in the absence of lymphocytes or external allergens, both in vivo and in vitro [127]. These findings highlight that TGFβ has a critical, irreplaceable role within epithelial cells in regulating tissue-specific allergic inflammation, independent of its function in adaptive immunity.

Recent studies have identified key molecular components involved in EoE fibrostenosis. In the study of fibrostenotic and non-fibrostenotic EoE phenotypes, researchers have identified a notable association between decreased TSPAN12 expression in endothelial cells and the onset of tissue fibrostenosis [128]. IL-13 reduced TSPAN12 expression, enhancing endothelial production of profibrotic mediators like endothelin-1 and boosting ECM production by fibroblasts [129].

Esophageal epithelial cells express IL-4α, IL-13Rα1, and IL-13Rα2, components of the IL-13 receptor, making them susceptible to IL-13. IL-13 plays a significant role in eotaxin-mediated eosinophil recruitment to the esophageal mucosa by modulating gene expression in epithelial cells. Both unaffected and EoE-affected esophageal tissues overexpress eotaxin-3 in response to IL-13, suggesting that the IL-13/IL-13 receptor/STAT6 pathway operates similarly in both groups [57]. Conversely, in epithelial cells stimulated by IL-13, cadherin-like 26, which regulates barrier function, is significantly upregulated [130]. The gene expression profiles of mucosal biopsies from EoE patients and primary esophageal epithelial cells treated with IL-13 are remarkably similar [131]. STAT6-dependent eotaxin-3 expression and secretion increase in primary esophageal epithelial cell cultures treated with IL-13 and in esophageal squamous carcinoma cell lines in an IL-13 dose-dependent manner [132]. IL-13-induced murine EoE is enhanced by deleting IL-13Rα2, suggesting that cytokine receptor subtypes may influence EoE expression. Genes associated with epithelial differentiation, such as filaggrin and SPRR3, are downregulated in EoE and primary epithelial cell cultures treated with IL-13 [57]. EoE patient biopsies treated with glucocorticoids show gene expression profiles similar to control individuals, as glucocorticoids inhibit IL-13-induced eotaxin-3 through the expression of FKBP51 [132].

The IL-20 subfamily (IL-19, IL-20, IL-24) significantly coordinates the immune system and epithelial cells. These cytokines activate type 1 and type 2 IL-20 receptors, leading to STAT3 phosphorylation and overlapping biological functions. In the skin, they induce keratinocyte changes, contributing to allergic skin and airway inflammation [133]. Our recent research revealed increased IL-20 subfamily cytokines in EoE patients’ esophagus and serum, downregulating genes and proteins of the cornified envelope, such as filaggrin. In *Il20R2^−/−^* mice, blocking IL-20 signaling alleviated EoE symptoms and maintained filaggrin expression through the MAPK/ERK1/2 pathway, preventing epithelial barrier impairment [134].

#### 7.3.2. IBD

The effect of cytokines on the intestinal epithelial barrier varies widely, with some cytokines consistently increasing permeability while others show controversial effects (extensively reviewed in [17]). IL-10 alone did not affect permeability in vitro [135] but effectively prevented barrier disruption induced by TNF or IFNγ [136,137]. IL-10 deficiency led to increased permeability under both normal and inflammatory conditions [138].

IL-13 increased epithelial permeability in vitro, indicated by decreased TEER and increased paracellular marker flux [139], and this effect was replicated in vivo in wild-type mice but not in STAT6-deficient mice [140]. IL-4 similarly increased gut permeability in vitro and in vivo, an effect prevented in IL-4Rα- and STAT6-deficient mice [140].

IL-1β consistently increased intestinal permeability in vitro and in vivo, partly by MLCK overexpression and miR200c-3p-mediated occludin degradation, with antagonism of miR200c-3p improving clinical outcomes in murine IBD models [141].

TNF, the most extensively studied cytokine, consistently increased permeability, especially in combination with IFNγ, which primes TNF receptors on cells [142]. In vivo, TNF administration increased permeability [143], and anti-TNF treatments reduced hyperpermeability in IBD models and patients [144].

Conversely, the effects of IL-17, IL-22, and IL-23 are more controversial. IL-17 showed mixed results: in vitro studies found both increased and decreased permeability, while in vivo studies indicated that IL-17 combined with IL-33 reduced permeability [145]. In IBD models, IL-17 inhibition increased permeability, yet it showed protective effects in other conditions [146]. IL-22 increased permeability in vitro but showed increased and decreased permeability in vivo, depending on the context. IL-23 increased permeability in vitro [147] but had opposite effects in vivo [146]. These findings highlight the complexity and context-dependence of cytokine effects on the intestinal barrier, necessitating further research to clarify these mechanisms for therapeutic purposes.

## 8. Comparative Immunopathogenesis of Eosinophilic Esophagitis and Inflammatory Bowel Disease

In this chapter, we will delve into the roles of eosinophils, T cells, B cells, mast cells, and myeloid cells. These immune cells are pivotal in the pathogenesis of EoE and play significant roles in IBD.

### 8.1. Eosinophils

#### 8.1.1. EoE

Eosinophils are the hallmark cells essential for diagnosing EoE, although their role in its pathogenesis remains unclear [148]. Clinical trials using anti-IL5 antibodies [149,150] have reduced eosinophilia but not improved clinical symptoms, indicating that other immune cells contribute to EoE. They contribute to tissue damage and remodeling by releasing toxic granule proteins, which directly damage epithelial cells, leading to inflammation and fibrosis [151]. Among these granule proteins, eosinophil peroxidase significantly impacts esophageal epithelial cells by inducing fibroblast growth factor 9 secretion, promoting basal cell hyperplasia [151]. Additionally, eosinophils produce TGF-β1, contributing to tissue remodeling and fibrosis [152]. Eosinophils secrete cytokines (e.g., IL-5, IL-13) and chemokines (e.g., eotaxins like CCL11, CCL24, CCL26) that recruit and activate more eosinophils and other inflammatory cells [153].

#### 8.1.2. IBD

Eosinophils are often found in increased numbers in the intestinal mucosa, particularly during active disease states. Their involvement is more pronounced in UC than in CD [154,155], where they contribute to mucosal inflammation and damage by releasing toxic granule proteins and cytokines [156], though to a lesser extent than in EoE. Eosinophil activation is driven by a broader range of cytokines, including IL-5, IL-13, IL-1β, and TNF [156]. In IBD, eosinophils amplify the inflammatory response and promote tissue remodeling and fibrosis. Additionally, eosinophils express TGF-β1, which has complex roles in inflammation and tissue remodeling, varying with the context [156].

### 8.2. T Cells

#### 8.2.1. EoE

Emerging evidence indicates that effector Th2 cells, which produce cytokines IL-4, IL-5, and IL-13, play a critical role in the pathogenesis of EoE. These cytokines promote eosinophilic inflammation, tissue remodeling, and fibrosis. IL-4 and IL-13 are essential for the activation and differentiation of Th2 cells and the subsequent recruitment of eosinophils and mast cells. Elevated numbers of CD4^+^ and CD8^+^ T cells and an increased CD8^+^ T cell/CD4^+^ T cell ratio in EoE biopsies [157]. In addition, mouse models of allergen-induced EoE have shown that CD4^+^ T cells are pathogenic, while CD8^+^ T cells play a less significant role [158]. Human tissue-resident CD3^+^ T cell analysis has identified a polyclonal memory CD4^+^ T cell subset expressing IL-4, IL-5, and IL-13, strongly associated with esophageal tissue eosinophilia [159].

Further characterization during active EoE versus remission has confirmed the presence of an enriched pathogenic memory effector Th2 cell population in the esophagus of individuals with active EoE. Additionally, Th2 cells exhibiting a CD154^+^ IL5^+^ phenotype have been detected in the peripheral blood of active EoE patients [160]. Notably, these peripheral T cells demonstrate reactivity to milk antigens in milk-induced EoE patients but not in control patients [160].

Regulatory T cells (Tregs) exhibit altered dynamics in EoE. Adult EoE patients typically display reduced Tregs in esophageal tissue [161]. In contrast, pediatric patients demonstrate a relative increase in Tregs [162], highlighting potential age-related differences in immune regulation in EoE.

#### 8.2.2. IBD

The immunopathogenesis of IBD diverges from that of EoE, with CD typically associated with a Th1/Th17 response. Interleukin-12, composed of interleukin-12p35 and interleukin-12p40 subunits, fosters Th1 cell differentiation, promoting IFNγ production and the recruitment of macrophages, natural killer cells and CD8^+^ T cells. In contrast, interleukin-6, TGF-β, and interleukin-1 drive the upregulation of interleukin-23R and transcription factors like retinoic acid-related orphan receptor gamma t (RORγt), facilitating Th17 cell differentiation. These cells produce interleukin-17A, interleukin-17F, and interleukin-22, fostering neutrophil recruitment and perpetuating inflammation. However, UC is more associated with a Th2 response, although it is not as exclusively Th2-dominated as EoE. Th2 cytokines (IL-4, IL-5, IL-13) and Tregs contribute to mucosal inflammation and tissue damage in UC. Tregs, producing anti-inflammatory cytokines like IL-10 and TGF-β, regulate the inflammatory response in both CD and UC. However, dysregulation of Tregs can lead to uncontrolled inflammation and tissue damage in IBD.

### 8.3. Mast Cells

#### 8.3.1. EoE

Mast cells are prominently found in the esophageal tissue of EoE patients and are critical players in the disease’s pathogenesis [124]. In active EoE, mast cells infiltrate and increase within the esophageal epithelium, where they become activated and degranulate [163]. They are in increased numbers in the esophageal epithelium and the lamina propria. Mast cells mediate inflammation and tissue remodeling by releasing histamine, proteases such as tryptase and chymase, and cytokines, including IL-13, IL-3, IL-5, and granulocyte-macrophage colony-stimulating factors, which activate eosinophils [163]. Additionally, they produce a range of other inflammatory mediators, including prostaglandins, leukotrienes, and thromboxanes, which enhance vascular permeability and cause smooth muscle contraction [164]. Their activation is often IgE-mediated, linking EoE to allergic responses. Mast cells interact with eosinophils and fibroblasts, promoting fibrosis and the characteristic esophageal remodeling seen in EoE [165]. They can also release TGF-β1, which contributes to tissue fibrosis and remodeling [165]. During homeostasis, a quiescent mast cell population resides in the lamina propria. However, during active disease, two additional mast cell populations emerge within the intraepithelial compartment, adopting a pro-inflammatory state and expressing proliferation-associated genes [163]. Notably, one of these populations persists even during disease remission, remaining ready to reinitiate inflammation [163]. This dynamic behavior underscores the critical role of mast cells in both the inflammatory and remodeling processes of EoE.

#### 8.3.2. IBD

In both UC and CD, there is an increase in mast cells within the intestinal mucosa, where they release pro-inflammatory mediators such as histamine, tryptase, TNFα, and IL-6 [166]. Mast cells in IBD interact with other immune cells, including T and dendritic cells, modulating the inflammatory response. Mast cells also promote epithelial barrier dysfunction by releasing mediators that disrupt tight junction proteins, thus increasing intestinal permeability [167]. Their role in IBD is multifaceted; they can both exacerbate and potentially regulate inflammation, depending on the cytokine environment and context [166].

### 8.4. B Cells

#### 8.4.1. EoE

The role of B cells in EoE is less well-defined than T cells and eosinophils, but they are still important. B cells are absent in mucosal biopsies from individuals without esophageal pathology. In EoE patients, the number of intraepithelial B cells correlates with mast cell numbers but not eosinophil counts [168]. Patients with EoE often have elevated serum IgE levels and increased sensitization to food and environmental allergens [168]. B cells contribute to the disease by producing IgG4 and IgE, which can bind to allergens and activate mast cells and eosinophils, perpetuating the inflammatory response. Experimental models of EoE, such as those involving intranasal administration of Aspergillus fumigatus, show a doubling of the resident B cell population [158]. Mice deficient in B cells exhibit significantly reduced eosinophil numbers in bronchoalveolar lavage fluid but not in the esophagus, suggesting a role for B cells in the disease, although not a critical one [158].

#### 8.4.2. IBD

In contrast to EoE, which is primarily driven by IgE-mediated mechanisms, the pathogenesis of IBD involves significant roles for both IgA and IgG. IgA is particularly important for mucosal immunity, aiding in maintaining intestinal barrier integrity and regulating microbial populations. IgA is transported across the epithelium in the intestine by FcRn receptors, enabling its protective functions within the gut’s mucosal environment. Conversely, in the esophagus, IgA is derived from the salivary glands, which play a crucial role in local immunity. While initial observations suggested that B cells might not be involved in IBD, subsequent studies revealed that rituximab, an anti-CD20 antibody, may not effectively target tissue-resident B cells, particularly antibody-producing plasma cells that express low levels of CD20 [169]. In inflamed IBD tissue, there is a notable predominance of IgG, contrasting with the IgA predominance observed in healthy gut tissue [170]. Therefore, this disparity raises the possibility that a deficiency in IgA, an increase in IgG, or both could contribute to the pathogenesis of IBD. Moreover, an elevation in commensal microbiota-specific IgG antibodies has been observed in the colonic mucosa of UC patients. In a murine model, induction of anti-commensal microbiota IgG antibodies resulted in intestinal inflammation through macrophage activation, recruitment of neutrophils, and the promotion of type 17 immunity, involving various immune cells producing type 17 cytokines such as ILC3s and Th17 cells [171].

### 8.5. Dendritic Cells and Macrophages

#### 8.5.1. EoE

In EoE, dendritic cells (DCs) are vital in initiating and perpetuating the type 2 inflammatory response. They act as antigen-presenting cells, capturing allergens and presenting them to naïve T cells, thereby driving their differentiation into Th2 cells [5]. These Th2 cells then produce IL-4, IL-5, and IL-13 cytokines. DCs in EoE are often activated by epithelial-derived cytokines like TSLP, which enhances their ability to prime Th2 responses [5]. Additionally, the presence of DCs in the esophageal mucosa correlates with disease activity, highlighting their importance in the local immune response [157]. In the esophageal mucosa, CD1a^+^ dendritic cells are found in low numbers. In children with EoE, CD1a^+^ cell numbers increase in the proximal esophagus but return to normal levels following fluticasone treatment, with no significant increase observed in the distal esophagus [102].

Research on macrophages in the esophagus remains limited. They are primarily studied in the context of esophageal cancers, where tumor-associated macrophages are noted [172]. However, their involvement in allergic diseases is significant. Macrophages activated by Th2 cytokines, particularly IL-4 and IL-13, adopt an M2 phenotype associated with wound healing and tissue repair. These alternatively activated macrophages produce extracellular matrix components and fibrogenic cytokines like TGF-β, contributing to the tissue remodeling and fibrosis observed in EoE.

#### 8.5.2. IBD

In IBD, DCs are involved in both the initiation and regulation of inflammation, but the immune response is more diverse, involving Th1, Th17, and regulatory T cell pathways (broadly covered in [173]). DCs in IBD capture microbial antigens from the gut lumen and present them to T cells, leading to the activation of Th1 and Th17 responses, which produce cytokines such as IFN-γ, IL-17, and TNF-α. These cytokines are critical in sustaining chronic intestinal inflammation and recruiting other immune cells like macrophages and neutrophils. DCs in IBD also play a role in maintaining the balance between pro-inflammatory and regulatory responses, with dysregulation contributing to the pathogenesis of the disease.

In IBD, macrophages play a dual role, with both pro-inflammatory and anti-inflammatory functions (explored in-depth in [174,175]). In CD, macrophages produce high levels of pro-inflammatory cytokines such as TNF-α, IL-1β, and IL-6, which drive chronic inflammation and tissue damage. These macrophages also participate in phagocytosis and the clearance of apoptotic cells and pathogens. In UC, macrophages contribute to the inflammatory milieu but also have roles in resolving inflammation and promoting mucosal healing. The balance between pro-inflammatory and anti-inflammatory macrophage activity is crucial in determining disease outcomes and response to therapy in IBD.

## 9. Summary and Conclusions

This comparative review delves into the intricate dynamics of immune–epithelium crosstalk in EoE and IBD. Despite affecting different parts of the gastrointestinal tract, both conditions share notable immunological and epithelial features, yet they exhibit distinct mechanisms and clinical manifestations. EoE primarily involves eosinophils and pathways like IL-13 and eotaxin-3, leading to esophageal remodeling and fibrosis. Conversely, IBD features a broader range of immune cells, including T cells, macrophages, and neutrophils, with diverse epithelial responses like ulceration and hyperplasia.

Both EoE and IBD involve significant interactions between epithelial cells and the immune system, where epithelial cells play a crucial role in sensing and responding to inflammatory stimuli. Cytokines secreted by immune cells profoundly affect the epithelium, influencing barrier integrity, cell proliferation, and immune responses. Both diseases demonstrate the importance of epithelial barrier integrity. In EoE, epithelial barrier defects are central to disease pathogenesis, allowing food antigen and allergen penetration across the epithelium and subsequent immune activation in the lamina propria. In IBD, barrier dysfunction invokes the invasion of microbes and their products through the epithelium, leading to chronic inflammation characterized by ulceration, epithelial hyperplasia, and dysplasia in different gut segments. We expect that a better understanding of cross-talks between immune cells and epithelial cells will lead to novel therapies in both diseases, as the healing of the epithelial barrier will protect the immune system from penetration of food antigens and microbes. The large family of G protein-coupled receptors (GPCRs) with >831 members are of particular interest as they are involved in the chemoattraction of immune cells and sense food- and microbial-derived metabolites. These receptors modulate epithelial barrier integrity and control cytokine production by immune cells that act on the epithelium. Most of these GPCRs are still orphans. Advancements in bioinformatic tools, such as AlphaFold2 and AlphaFold3, will reveal the tertiary structures of these GPCRs and provide indications of potential ligands through molecular dynamic simulations, leading to novel approaches for treating EoE and IBD.

In conclusion, while EoE and IBD are distinct diseases with unique pathophysiological mechanisms, they share fundamental processes involving the immune–epithelium crosstalk, which results in an impaired epithelial barrier in both entities. A deeper understanding of these interactions holds promise for advancing the diagnosis, treatment, and management of these chronic inflammatory conditions.

## Figures and Tables

**Figure 1 ijms-25-08534-f001:**
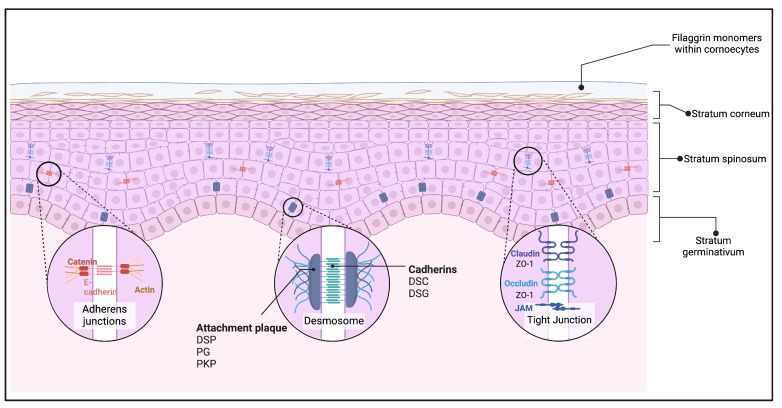
**Schematic representation of healthy esophageal epithelium.** This figure illustrates the typical architecture of the human esophageal squamous epithelium. It shows the basal layer with actively proliferating cells, the spinous layer where cells begin to differentiate and flatten, and the superficial layer where fully differentiated cells form a protective barrier. Essential proteins such as E-cadherin, claudin-1, and desmoglein-1 are indicated to highlight their roles in maintaining barrier integrity and cell–cell junctions. These proteins fuse cells in the stratum spinosum by forming the junction complex, which consists of the tight junction complex, the adherens junction complex, and the desmosomes, with the tight junction complex located most apically. The healthy epithelium protects against external antigens, maintaining proper function and barrier integrity. Created with Biorender.com.

**Figure 2 ijms-25-08534-f002:**
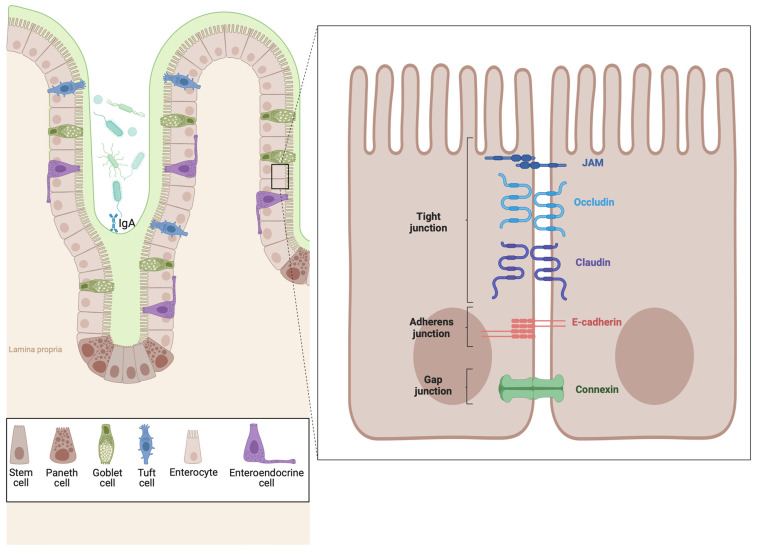
**Schematic representation of healthy intestinal epithelium.** This figure illustrates the architecture of the healthy intestinal epithelium. It highlights various cell types, including enterocytes, goblet, and Paneth cells, which maintain the intestinal barrier. Enterocytes form three types of junctions: tight junctions, which seal the space between cells to regulate permeability; adherens junctions, which provide mechanical stability by linking the actin cytoskeletons of neighboring cells; and desmosomes, which offer additional strength by anchoring intermediate filaments. Goblet cells secrete mucus to protect the epithelial surface, while Paneth cells at the crypt bases produce antimicrobial peptides. Microvilli on enterocytes increase the surface area for nutrient absorption, ensuring efficient digestion and immune protection. Created with Biorender.com.

**Table 1 ijms-25-08534-t001:** **Treatment strategies for EoE and IBD**. It highlights EoE’s emphasis on topical corticoids and dietary elimination. At the same time, IBD management includes a broader array of treatments, such as biologics and surgical options, tailored to disease severity and patient response.

Aspect	EoE	IBD
**Main Treatment Focus**	Reduce esophageal inflammation to alleviate symptoms and prevent complications such as strictures	Induce and maintain remission to alleviate symptoms, enhance quality of life, and prevent complications
**Dietary Modifications**	One- or six food elimination diet (identifying and eliminating specific food triggers)	Not commonly used as primary treatment
**Medications**	PPIs to suppress gastric acid and reduce inflammation; swallowed topical corticosteroids for symptomatic relief and mucosal healing; the IL-4Ra antibody dupilumab targeting IL-13 and IL-4/IL-13	5-ASAs for mild-to-moderate UC; corticosteroids for acute flares; immunomodulators (azathioprine, 6-mercaptopurine, methotrexate) for maintenance therapy; biologics targeting TNF-α, interleukins, integrins (e.g., anti-TNF agents, anti-integrins, anti-IL-12/23 or anti-IL-23 agents); JAK inhibitors
**FMT**	Not applicable	Restoring microbiome diversity
**Surgical Interventions**	Reserved for complications, such as perforations	Bowel resection or ostomy for complications like strictures, perforations, or refractory disease unresponsive to medical therapy

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
