# Peer review of "Eosinophilic Esophagitis and Inflammatory Bowel Disease: What Are the Differences?"

_ijms, 2024, doi:10.3390/ijms25158534_

Round 1

Reviewer 1 Report

Comments and Suggestions for Authors

This represents a thorough manuscript dealing with the similarities and differences between eosinophilic esophagitis (EoE) and different forms of inflammatory bowel diseases inclusive of Crohn's disease and ulcerative colitis which have overlapping features while largely impacting different segments of the gastrointestinal tract.

Minor comments:

Line 146: Is there supposed to be a "." like this: "regions. GWAS" rather than "regions GWAS"

Line 146: There are "five" loci listed rather than the "four" in the text.

Line 275: Reference 86 doesn't appear to be appropriate here as it predates the approvals of dupilumab for the treatment of EoE in the USA and Europe.

Line 429: Refence 114 relates to colitis rather than EoE

Line 431: Refence 114 relates to colitis rather than EoE

Author Response

Reviewer 1

This represents a thorough manuscript dealing with the similarities and differences between eosinophilic esophagitis (EoE) and different forms of inflammatory bowel diseases, including Crohn's disease and ulcerative colitis, which have overlapping features while largely impacting different segments of the gastrointestinal tract.

Authors: We thank Reviewer 1 for his/her comments.

Minor comments:

Line 146: Is there supposed to be a "." like this: "regions. GWAS" rather than "regions GWAS"?

Authors: We apologize for this oversight. We have now inserted a "." between "regions" and "GWAS" (Refer to line 206 in the revised review).

Line 146: There are "five" loci listed rather than the "four" in the text.

Authors: We acknowledge this discrepancy. We have corrected the text to read "five" instead of "four" (Refer to line 206 in the revised review)

Line 275: Reference 86 doesn't appear to be appropriate here as it predates the approvals of dupilumab for the treatment of EoE in the USA and Europe.

Authors: We agree with the reviewer that reference 86 predates the approvals of dupilumab for treating EoE in the USA and Europe. However, since the receptors of both IL-4 and IL-13 share the same alpha chain and dupilumab targets this shared alpha chain, we believe the reference remains relevant and cite now the Dellon New England Manuscript (Refer to reference 89, lines 334-336 in the revised review).

Line 429: Reference 114 relates to colitis rather than EoE.

Authors: We agree with the reviewer. We have removed reference 114 and replaced it with reference 124 (Refer to line 517 in the revised review).

Line 431: Reference 114 relates to colitis rather than EoE.

Authors: We agree. We have removed reference 114 ,and replaced it with reference 124, and added reference 125 (Refer to line 517 in the revised review).

Reviewer 2 Report

Comments and Suggestions for Authors

In this review, the authors attempt to provide an overview of eosinophilic esophagitis and IBD, discussing their commonalities and differences. However, the structure of the review lacks cohesion. My major comments are as follows:

  1. The abstract does not effectively summarize the main points of the review. Instead, it focuses on several minor aspects, such as GPCR, autophagy, and specific immune cells.
  2. Lines 118-119: The statistics on IBD prevalence are incorrect. The authors should verify the reference and cite the most updated statistics.
  3. The summary of IBD treatment does not mention fecal microbiota transplantation, which is clinically used and should be included.
  4. Lines 138-140: The dataset cited is very small and lacks population diversity. Additionally, some statistics in the reference are incorrect. The authors should verify the data and compare it with other similar prevalence reports.
  5. Line 645: The statement that 'macrophages play a limited role in the esophagus' is unconvincing.
  6. Line 715: There is a typo in '...adress this gap knowledge...' which should be corrected to '...address this gap in knowledge...'.
  7. The discussion on GPCR and autophagy in the pathogenesis and treatment potential of the two diseases is too general and superficial, making this section appear disorganized.
  8. There is no summary section in the review.

Author Response

Reviewer 2

In this review, the authors attempt to provide an overview of eosinophilic esophagitis and IBD, discussing their commonalities and differences. However, the structure of the review lacks cohesion. My major comments are as follows:

Authors: We thank Reviewer 2 for his/her comment. We have revised the abstract and the summary / conclusions to improve the cohesion of the manuscript.

The abstract does not effectively summarize the main points of the review. Instead, it focuses on several minor aspects, such as GPCR, autophagy, and specific immune cells.

Authors: The reviewer raises a valid point. We have revised the abstract to better reflect the entire review structure. Additionally, we removed the discussion of autophagy from both the abstract and the discussion section, focusing instead on the role of GPCRs in EoE as a future research direction. GPCRs may play a significant role in EoE pathogenesis by sensing metabolites derived from food implicated in the disease or by regulating the cytokine milieu that affects the esophageal epithelium (Refer to the revised abstract, lines 10-26 in the revised review).

Lines 118-119: The statistics on IBD prevalence are incorrect. The authors should verify the reference and cite the most updated statistics.

Authors: We appreciate this observation. We corrected the prevalence of IBD in the European population. Additionally, as suggested by the reviewer, we have replaced the reference for IBD prevalence in the USA with a more recent source (Refer to lines 163-176, Reference 34 in the revised review).

The summary of IBD treatment does not mention fecal microbiota transplantation, which is clinically used and should be included.

Authors: Following the reviewer's suggestion, we have now included fecal microbiota transplantation in the IBD treatment section (Refer to lines 348-352 and Table 1 in the revised review). 

Lines 138-140: The dataset cited is very small and lacks population diversity. Additionally, some statistics in the reference are incorrect. The authors should verify the data and compare it with other similar prevalence reports.

Authors: We acknowledge this concern. We have replaced the dataset with a larger one, as recommended by the reviewer (Refer to lines 189 and 190, Reference 41 in the revised review).

Line 645: The statement that 'macrophages play a limited role in the esophagus' is unconvincing.

Authors: We agree with the reviewer. We have amended the sentence to "Research on macrophages in the esophagus remains limited” (Refer to line 733 in the revised review).

Line 715: There is a typo in '...adress this gap knowledge...' which should be corrected to '...address this gap in knowledge...'.

Authors: We apologize for this typo. We have revised the entire paragraph (Refer to our latest response answer below)

The discussion on GPCR and autophagy in the pathogenesis and treatment potential of the two diseases is too general and superficial, making this section appear disorganized.

Authors: We appreciate this feedback. We have amended this section following the reviewer’s suggestion (Refer to our latest response anser below).

There is no summary section in the review.

Authors: We have revised the last section of the review, now titled "Summary and Conclusion," to include a summary of the key sections and a paragraph on future directions for EoE therapeutics. (Refer to the  Summary and Conclusion section in the revised review, lines 759-797).

Round 2

Reviewer 2 Report

Comments and Suggestions for Authors

The authors solved my concerns and I have no further comment.

Author Response

We thank the reviewer for the kind comments.